# Enhancing Low-Pass Filtering Detection on Small Digital Images Using Hybrid Deep Learning

**Saurabh Agarwal** [1,2] and **Ki-Hyun Jung** [1,*]

1. Department of Software Convergence, Andong National University, Andong 36729, Gyeongbuk, Republic of Korea; saurabhnsit2510@gmail.com
2. Amity School of Engineering & Technology, Amity University Uttar Pradesh, Noida 201313, India
* Correspondence: kingjung@anu.ac.kr; Tel.: +82-54-820-7968

**Abstract:** Detecting image manipulation is essential for investigating the processing history of digital images. In this paper, a novel scheme is proposed to detect the use of low-pass filters in image processing. A new convolutional neural network with a reasonable size was designed to identify three types of low-pass filters. The learning experiences of the three solvers were combined to enhance the detection ability of the proposed approach. Global pooling layers were employed to protect the information loss between the convolutional layers, and a new global variance pooling layer was introduced to improve detection accuracy. The extracted features from the convolutional neural network were mapped to the frequency domain to enrich the feature set. A leaky Rectified Linear Unit (ReLU) layer was discovered to perform better than the traditional ReLU layer. A tri-layered neural network classifier was employed to classify low-pass filters with various parameters into two, four, and ten classes. As detecting low-pass filtering is relatively easy on large-dimension images, the experimental environment was restricted to small images of $30 \times 30$ and $60 \times 60$ pixels. The proposed scheme achieved 80.12% and 90.65% detection accuracy on ten categories of images compressed with JPEG and a quality factor 75 on $30 \times 30$ and $60 \times 60$ images, respectively.

**Keywords:** low-pass filtering detection; image smoothing detection; image forgery detection; image manipulation detection; fake image; image forensics

## 1. Introduction

Digital images are a commonly used medium, especially for informal information sharing. However, image contents can be changed without difficulty by using specific software. Usually, resizing [1], gamma correction [2], sharpening [3], and smoothing [4] operations are used to produce a convincing fake image. Image forensics provides the details of past operations applied to the image. This paper's primary objective was to detect the low-pass filter operation usually used for smoothing. Three commonly used low-pass filters—Gaussian, median, and averaging—are addressed in this paper. The images displayed in Figure 1 show the visual effect on the image after low-pass filtering and JPEG compression. Uncompressed images are displayed in the first row, and their compressed version of 75 quality factor in the second row. The structural similarity index measure (SSIM) [5] relates to visual quality. The first image in the first row is the reference image, and its assumed SSIM quality value (SQ) is 1.0. In the first row, the second, third, and fourth images are averaging- (SQ = 0.48), Gaussian- (SQ = 0.92), and median-filtered (SQ = 0.52) images, respectively. The Gaussian-filtered image is created using a Gaussian filter with a standard deviation of 0.5 and a filter window size of $3 \times 3$. The SQ is highest for the Gaussian-filtered image after the original image. The SQ is related to visual quality. In the second row, compressed images are displayed with their corresponding SQs. The SQ has dropped greatly for a compressed Gaussian-filtered image compared to other filtered images. However, the SQ value of the compressed Gaussian-filtered image is highest in the compressed filtered images.

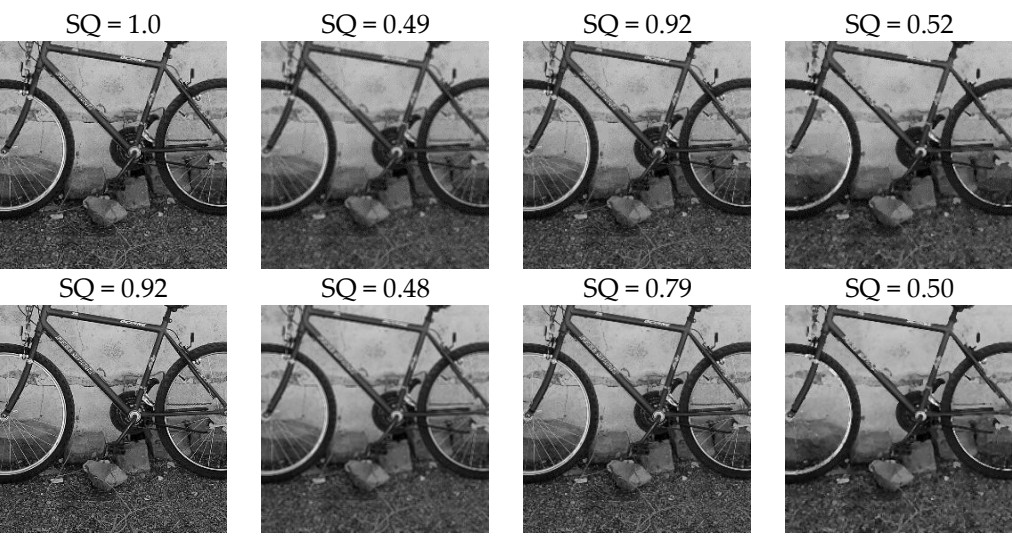

**Figure 1.** Uncompressed and compressed low-pass-filtered images.

The means of SSIM quality values (SQ) are displayed in Figure 2 to observe the general behavior of filtered images. The low-pass filters were applied to one thousand images with dimensions $30 \times 30$ and $60 \times 60$. ORI denotes the non-filtered images. GA1, GA2, and GA3 indicate the Gaussian-filtered images of filter sizes $3 \times 3$, $5 \times 5$, and $7 \times 7$. Likewise, AG represents averaging-filtered images, and MD denotes median-filtered images. JPEG-compressed images are also considered with quality factor (JQF) 75. It can be observed from the figure that as the filter size is enlarged, the SQ value declines. The SQ values of GA are followed by those for AG and MD.

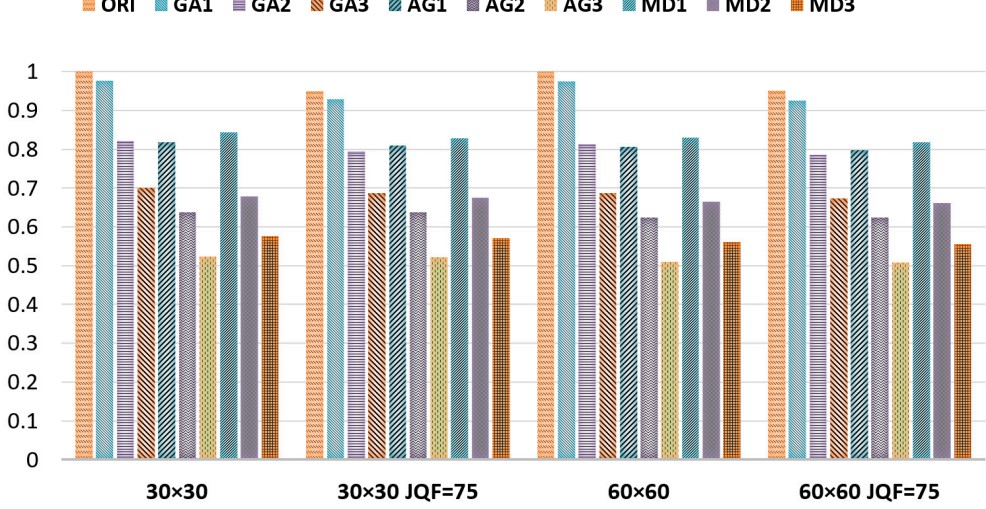

**Figure 2.** SQ values for uncompressed and compressed images.

One more image quality indicator—perception-based image quality evaluator (PIQUE) [6]—is considered for analysis. PIQUE is a no-reference quality metric for blind image quality assessment. The covariance plot of the PIQUE score is displayed in Figure 3 to reveal the difference in nature of non-filtered, low-pass-filtered images for all ten classes, including three low-pass filters with different parameters. In Figure 3a, uncompressed images are considered; in Figure 3b, compressed images with JQF = 75 are considered to evaluate the PIQUE score.

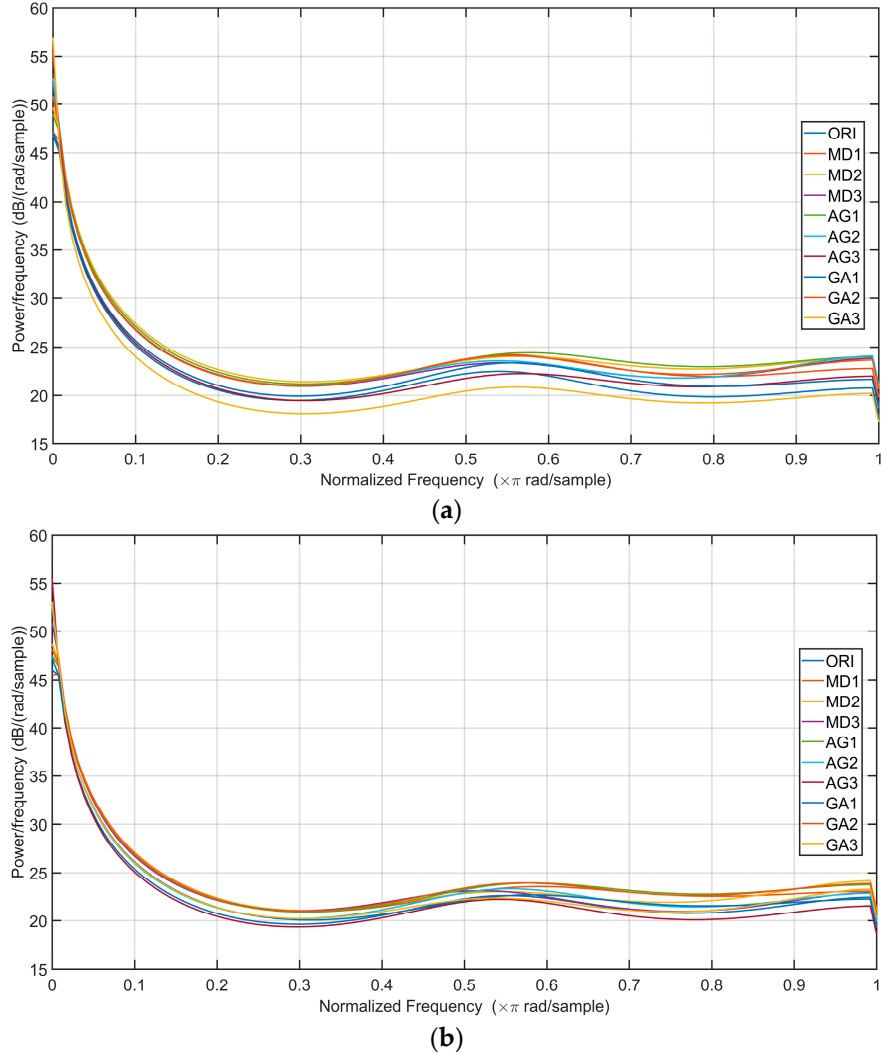

**Figure 3.** Covariance plot of PIQUE score on ten categories of images. (**a**) Uncompressed images; (**b**) compressed images.

As discussed above, the low-pass-filtered images with low-potency filters, such as filter size 3 × 3, have less difference in SQ compared to non-filtered images. Researchers suggest several schemes to address the issue of low-pass filtering. Kirchner and Fridrich [7] detected median filtering by measuring the co-occurrence between pixels. Similarly, Chen et al. [8] and Agarwal & Jung [4] measured the co-occurrence of pixels using some additional modifications. Chen et al. [8] also added local features by calculating local correlation. Agarwal and Jung [4] utilized the higher-significance bit-plane to improve the detection capability on low-pass-filtered images. Gupta and Singhal [9] exploited the discrete cosine domain to extract four features of first-order moments. The autoregressive model detects filtering in schemes [10–12]. Kang et al. [10] considered the model of order ten to detect median filtering. Yang et al. [11] applied the 2D model on several types of difference arrays for better results. Rhee [12] considered the bit planes to boost performance using the autoregressive model. Chen et al. [13] introduced the deep neural network to recognize median-filtered images. The network had five convolutional layers, and median-filter residual images were considered for training. The deep neural network of Liu et al. [14] utilized the discrete Fourier transform domain images to detect low-pass filtering. The network had merely two convolutional layers and pooling layers. The Zhang et al. [15] deep neural network contained squeeze and excitation blocks to detect three types of low-pass filtering. The computational cost of the Zhang et al. network is expensive. Yu et al. [16] preprocessed the images with several high-pass filters before sending them to a deep neural

network with four convolutional layers. The co-occurrence and autoregressive model extracted the prevailing manual scheme features. In the prevailing convolutional neural network (CNN)-based schemes, altered arrangements of layers are performed to improve the performance. However, deep neural network schemes provide better results than the prevailing manual schemes. In this paper, a hybrid scheme is proposed that has a deep neural network at its core. The proposed scheme has superior recognition competency and reasonable computational expense. The key highlights of the proposed scheme are as follows:

- The proposed scheme is suitable for detecting various types and strengths of low-pass filters. The scheme is suitable for large- and small-dimension uncompressed and JPEG-compressed images.
- The convolutional neural network of the proposed scheme comprises merely nine convolutional layers with a lesser number of kernels.
- The global variance pooling layer is introduced to improve the detection performance of the deep neural network of the proposed scheme.
- Feature arrays from trained networks are combined to boost detection accuracy. The network is trained using three different solvers.
- The spatial domain and discrete Fourier transform domain features are concatenated to enhance the potential of the proposed scheme.
- A leaky ReLU layer is employed instead of a conventional ReLU layer to optimize the results.
- A tri-layered neural network classifier is employed to classify a different set of images that are processed by low-pass filters.

The remainder of the paper is structured as follows: Section 2 describes the suggested technique. Multiple features are fetched in the proposed scheme using two global pooling layers and three optimizers. Spatial domain features are also mapped in the frequency domain. Both domain features are concatenated for better results. The experiment's findings for the proposed scheme and the paper's conclusions are discussed in Sections 3 and 4, respectively.

## 2. Proposed Scheme

Low-pass filtering is applied for smoothing and blurring the image. In this paper, a robust scheme is suggested to detect low-pass filtering. The proposed scheme incorporates the features of spatial and frequency domains by using three different solvers in a convolutional neural network. First, the proposed deep neural network is explained. It is found from the regressive experimental analysis that for low-pass filtering detection, a light neural network provides better results than a CNN that has residual connections and a high number of parameters. The ResNet [17] and other computationally expensive CNNs do not provide satisfactory detection performance. Figure 4 presents a proposed deep neural network containing nine two-dimensional (2D) convolutional (Conv) layers. A batch normalization (Batch Norm) layer and a leaky ReLU layer trail each 2D convolutional layer. A leaky ReLU layer is used instead of the conventional ReLU layer in the proposed convolutional neural network to improve detection capability. In a leaky ReLU layer, every negative element is multiplied by a constant real positive number ($c$). The elements greater than or equivalent to zero stay the same. The leaky ReLU layer ($LR$) can be defined as follows:

$$LR(n) = \begin{cases} n, & n \geq 0 \\ c \times n, & n < 0 \end{cases}$$

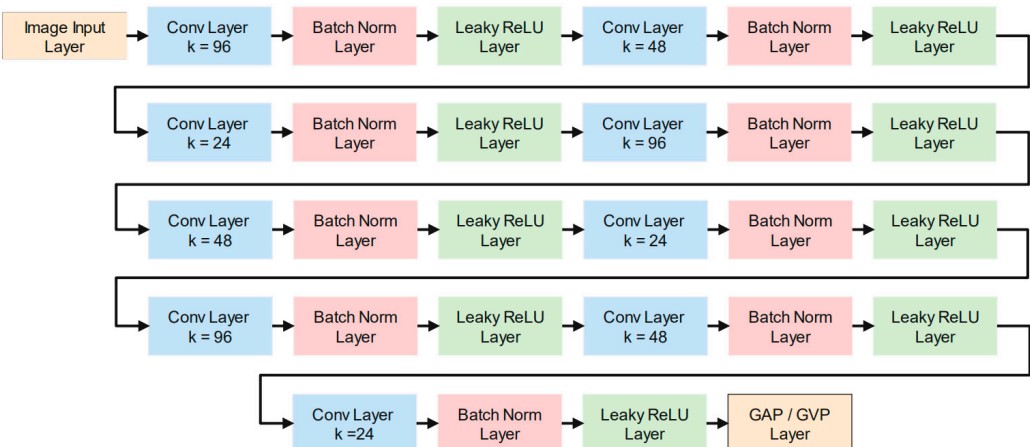

**Figure 4.** Design of the proposed convolutional neural network.

However, every negative element is assigned a zero value, and positive elements remain unchanged in the conventional ReLU layer. ReLU layer (*R*) can be defined as follows:

$$R(n) = \begin{cases} n, & n \geq 0 \\ 0, & n < 0 \end{cases}$$

There are 96, 48, 24, 96, 48, 24, 96, 48, and 24 kernels in the head 2D convolutional layer to the last convolutional layer. Several combinations of layers and the number of kernels were tried. However, the suggested convolutional neural network was found more suitable. No other pooling layers were used in the network to protect against the loss of information between the convolutional layers. The global pooling layers were used to amplify the performance. Global average pooling (GAP) and global variance pooling (GVP) layers were considered.

Subsequently, in the last leaky ReLU layer, global pooling layers were used to boost the detection capability. In some previous schemes [18,19], the global average pooling layer enhanced the stego image detection accuracy. The proposed scheme also introduces a global variance pooling layer to make the robust feature array. Figure 5 displays layouts of the global average and variance pooling layers. Instead of calculating the average of the feature map, variance is calculated in the global variance pooling layer.

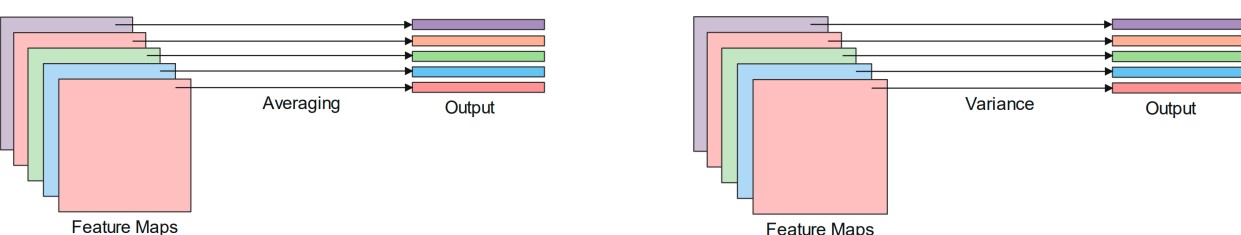

**Figure 5.** GAP and GVP layers.

The CNN was trained twice, once using the GAP layer and once using the GVP layer. Further, an activation function was applied, as displayed in Figure 6, to extract the feature arrays. The feature arrays i and j were concatenated. The variance provides the additional details of low-pass-filtering traces that assist in distinguishing non-filtered and filtered images.

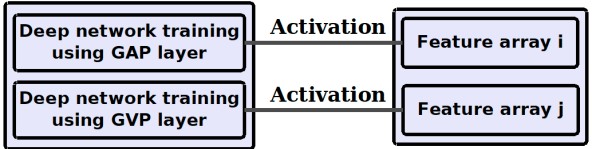

**Figure 6.** Features fetched while using GAP and GVP layer.

Three different solvers were used in network training to strengthen the feature array. These solvers are adaptive moment (Adam), root mean square propagation (RMSprop), and stochastic gradient descent with moment (Sgdm). The activation function was applied to the trained network, as displayed in Figure 7. Feature arrays were fetched from the trained networks to acquire the benefit of three solvers. Each feature array was also converted into the DFT domain. The conversion of features in the DFT domain is beneficial in steganalysis by Wang et al. [20]. This fact is also found relevant in the experimental analysis of our work. This process created six feature arrays. In the proposed CNN, two types of global pooling layers were used, one layer at a time. Therefore, a total of twelve feature arrays were formed. These twelve feature arrays were concatenated to generate a single feature array.

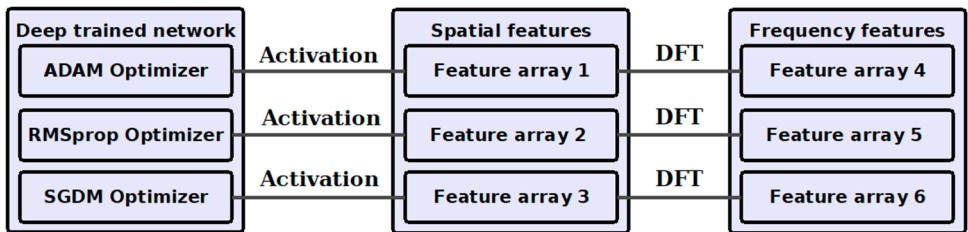

**Figure 7.** Features fetched from three different solvers and converted into the DFT domain.

Further, a tri-layered neural network classifier was applied to feature arrays to classify non-filtered and low-pass-filtered images. A tri-layered neural network classifier provided better results than softmax, support vector machine, and linear discriminant analysis classifiers in our experimental analysis. Fifty epochs were engaged to train the network. Mini batch sizes of 300 and 150 were deliberated separately for images with dimensions $60 \times 60$ and $30 \times 30$. The proposed work was performed on a system with the following hardware details: NVIDIA GeForce RTX3070, Intel(R) Core (TM) i7-10700K CPU @ 3.80 GHz, and 32 GB RAM. The network learning duration is displayed in Table 1 with Adam, RMSprop, and Sgdm solvers for ten class categorizations. In the case of uncompressed images, learning time is compared to compressed images due to more information content.

**Table 1.** CNN learning duration in minutes.

| Image Dimensions/Solver | Adam | RMSprop | Sgdm |
|:---:|:---:|:---:|:---:|
| $30 \times 30$ | 121.12 | 104.67 | 125.92 |
| $60 \times 60$ | 198.29 | 213.81 | 219.35 |
| $30 \times 30$ (JQF = 75) | 52.61 | 53.62 | 64.88 |
| $60 \times 60$ (JQF = 75) | 163.86 | 170.92 | 154.16 |

In Figure 8, feature arrays are displayed for four classes of non-filtered images and median-filtered images that were processed with filter sizes $3 \times 3$, $5 \times 5$, and $7 \times 7$. The first plot shows the feature arrays of the uncompressed images with dimensions $30 \times 30$, and the second plot displays the feature arrays of the compressed images with JQF = 75 and size $30 \times 30$. It can be perceived from Figure 8 that feature arrays of uncompressed images are more distinguishable than compressed images. This distinguishability of feature arrays is also reflected in the experimental analysis, indicating that the detection performance on uncompressed images is superior to compressed images.

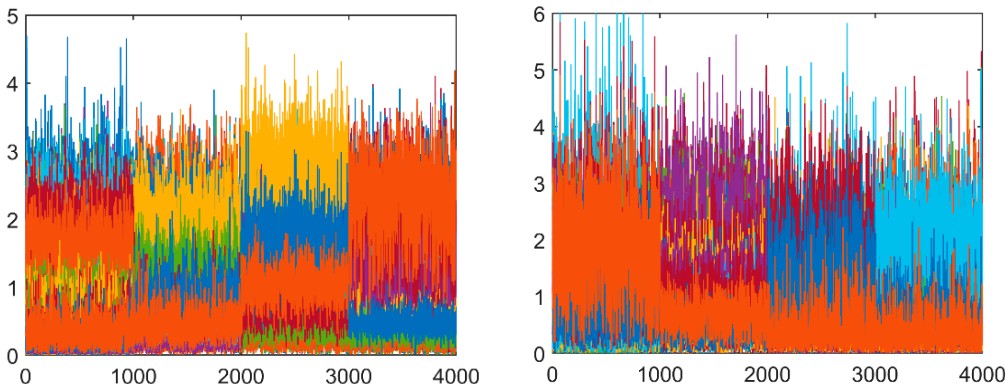

**Figure 8.** Feature arrays of uncompressed and compressed images for the median filter.

## 3. Experimental Analysis and Discussion

This paper proposes a scheme to detect low-pass filtering in images. An image set was created from three different image datasets, i.e., UCID [21], BOSSBase 1.01 [22], and NRCS [23], to assess the detection scheme. UCID dataset comprises 1338 uncompressed color images with dimensions 512 × 384. The BOSSBase 1.01 dataset has ten thousand uncompressed grayscale images with dimensions 512 × 512. Two thousand high-resolution images were taken from an NRCS photo gallery containing landscapes, soil, etc. Thirty-five thousand images of size 60 × 60 were generated by cropping images in random order. The images with dimensions 30 × 30 were generated by selecting an interior chunk of images with dimensions 60 × 60. Respective low-pass filtered images were generated by applying a particularly low-pass filter. In the case of training the network, twenty thousand images in the required category were taken. Fifteen thousand images were employed for training. Uncompressed and JPEG-compressed images of quality factor 75 were considered for rigorous experimental analysis. JPEG compression was applied after filtering.

The proposed scheme is denoted as Z1, Z2, and Z3 while using Sgdm, Adam, and RMSprop, respectively, as solvers. ZALL represents the final scheme by concatenating the feature arrays of Z1, Z2, and Z3. ORI stands for unfiltered images. Both spatial and frequency domain features were concatenated for better results. The Gaussian-filtered images of sizes 3 × 3, 5 × 5, and 7 × 7 are designated as GA1, GA2, and GA3, respectively. In the same way, AG stands for averaging-filtered images, and MD for median-filtered images. Two, four, and ten categories were classified using a tri-layered neural network classifier for in-depth analysis.

Tables 2 and 3 display the detection accuracy of two-class classification. The detection accuracy of the four-class classification is shown in Tables 4 and 5. The detection accuracy of the ten-class classification is given in Tables 6 and 7.

**Table 2.** Two-class classification for 30 × 30 uncompressed images.

| Image Pair | Image Dimensions 30 × 30 | | | | | | | | |
| --- | --- | --- | --- | --- | --- | --- | --- | --- | --- |
| | ORI-GA1 | ORI-GA2 | ORI-GA3 | ORI-AG1 | ORI-AG2 | ORI-AG3 | ORI-MD1 | ORI-MD2 | ORI-MD3 |
| Z1 | 83.71 | 99.74 | 99.94 | 99.94 | 99.69 | 99.86 | 98.97 | 99.34 | 99.76 |
| Z2 | 83.68 | 99.71 | 99.91 | 99.90 | 99.67 | 99.83 | 98.94 | 99.32 | 99.73 |
| Z3 | 83.70 | 99.73 | 99.94 | 99.92 | 99.70 | 99.32 | 98.54 | 99.35 | 99.65 |
| ZALL | 84.72 | 99.74 | 99.93 | 99.92 | 99.70 | 99.87 | 99.87 | 99.34 | 99.75 |

**Table 3.** Two-class classification of compressed images.

| | | | | Image Dimensions 30 × 30 JQF = 75 | | | | |
|---|---|---|---|---|---|---|---|---|
| Image Pair | ORI-GA1 | ORI-GA2 | ORI-GA3 | ORI-AG1 | ORI-AG2 | ORI-AG3 | ORI-MD1 | ORI-MD2 | ORI-MD3 |
| Z1 | 72.40 | 95.25 | 98.96 | 95.70 | 99.07 | 99.28 | 91.76 | 97.65 | 98.17 |
| Z2 | 73.00 | 95.23 | 98.78 | 96.12 | 99.26 | 99.11 | 91.67 | 97.92 | 97.65 |
| Z3 | 72.07 | 95.75 | 98.38 | 95.70 | 98.83 | 99.30 | 91.95 | 98.03 | 98.52 |
| ZALL | 74.07 | 96.26 | 99.32 | 96.49 | 99.39 | 99.45 | 93.06 | 98.87 | 98.70 |
| | | | | Image dimensions 60 × 60 JQF = 75 | | | | |
| Image pair | ORI-GA1 | ORI-GA2 | ORI-GA3 | ORI-AG1 | ORI-AG2 | ORI-AG3 | ORI-MD1 | ORI-MD2 | ORI-MD3 |
| Z1 | 80.53 | 98.15 | 99.60 | 97.63 | 99.81 | 99.63 | 97.10 | 99.32 | 99.56 |
| Z2 | 80.43 | 97.91 | 99.46 | 97.48 | 99.57 | 99.52 | 96.95 | 99.17 | 99.85 |
| Z3 | 80.58 | 98.11 | 99.74 | 97.64 | 99.73 | 99.74 | 97.24 | 99.32 | 99.68 |
| ZALL | 81.63 | 98.89 | 99.83 | 98.45 | 99.74 | 99.79 | 98.23 | 99.45 | 99.98 |

**Table 4.** Four-class classification on uncompressed images for Gaussian filtering.

| Class | Image Dimensions 30 × 30 | | | | | Image Dimensions 60 × 60 | | | | |
|---|---|---|---|---|---|---|---|---|---|---|
| | ORI | GA1 | GA2 | GA3 | Overall | ORI | GA1 | GA2 | GA3 | Overall |
| Z1 | 87.21 | 92.03 | 97.29 | 99.14 | 93.92 | 94.86 | 96.67 | 99.19 | 99.49 | 97.55 |
| Z2 | 88.25 | 93.29 | 95.69 | 99.63 | 94.21 | 96.09 | 97.23 | 98.68 | 99.83 | 97.96 |
| Z3 | 90.75 | 89.67 | 97.28 | 99.49 | 94.30 | 95.31 | 96.55 | 99.33 | 99.50 | 97.67 |
| ZALL | 92.20 | 92.06 | 98.99 | 99.61 | 95.71 | 97.50 | 98.22 | 99.96 | 99.73 | 98.85 |

**Table 5.** Four-class classification of compressed images.

| Image Dimensions 30 × 30 JQF = 75 | | | | | Image Dimensions 60 × 60 JQF = 75 | | | | |
|---|---|---|---|---|---|---|---|---|---|
| Class | ORI | GA1 | GA2 | GA3 | Overall | Class | ORI | GA1 | GA2 | GA3 | Overall |
| Z1 | 62.05 | 71.91 | 85.65 | 97.50 | 79.28 | Z1 | 77.40 | 83.25 | 89.91 | 99.50 | 87.51 |
| Z2 | 68.63 | 68.92 | 85.60 | 94.86 | 79.50 | Z2 | 69.66 | 88.28 | 90.35 | 99.39 | 86.92 |
| Z3 | 64.75 | 73.98 | 86.47 | 94.83 | 80.01 | Z3 | 77.59 | 83.17 | 91.97 | 99.26 | 87.99 |
| ZALL | 70.47 | 71.47 | 86.87 | 97.69 | 81.62 | ZALL | 81.28 | 83.53 | 93.54 | 98.65 | 89.25 |
| Class | ORI | AG1 | AG2 | AG3 | Overall | Class | ORI | AG1 | AG2 | AG3 | Overall |
| Z1 | 93.27 | 94.86 | 97.40 | 97.31 | 95.71 | Z1 | 97.63 | 97.81 | 98.40 | 99.55 | 98.34 |
| Z2 | 92.11 | 94.49 | 96.09 | 97.65 | 95.09 | Z2 | 97.33 | 97.85 | 98.69 | 99.42 | 98.32 |
| Z3 | 93.63 | 95.06 | 96.23 | 98.11 | 95.76 | Z3 | 97.03 | 98.35 | 97.99 | 99.69 | 98.26 |
| ZALL | 94.99 | 95.79 | 97.83 | 98.80 | 96.85 | ZALL | 98.69 | 98.86 | 98.91 | 99.61 | 99.02 |
| Class | ORI | MD1 | MD2 | MD3 | Overall | Class | ORI | MD1 | MD2 | MD3 | Overall |
| Z1 | 90.06 | 89.39 | 92.66 | 89.91 | 90.51 | Z1 | 94.25 | 97.01 | 96.71 | 97.17 | 96.29 |
| Z2 | 86.64 | 91.10 | 90.85 | 94.23 | 90.71 | Z2 | 95.73 | 96.31 | 96.35 | 98.22 | 96.65 |
| Z3 | 91.99 | 85.66 | 91.26 | 94.31 | 90.80 | Z3 | 96.79 | 95.56 | 96.53 | 98.13 | 96.75 |
| ZALL | 90.77 | 90.20 | 91.51 | 95.89 | 92.09 | ZALL | 96.83 | 97.67 | 97.75 | 98.99 | 97.81 |

**Table 6.** Ten-class classification of uncompressed images.

| Class | ORI | MD1 | MD2 | MD3 | GA1 | GA2 | GA3 | AG1 | AG2 | AG3 | Overall |
|---|---|---|---|---|---|---|---|---|---|---|---|
| | | | | Image Dimensions 30 × 30 | | | | | | | |
| Z1 | 91.84 | 99.03 | 97.11 | 98.53 | 85.61 | 97.82 | 98.07 | 94.67 | 99.27 | 99.43 | 96.14 |
| Z2 | 91.52 | 99.09 | 98.48 | 97.15 | 93.85 | 97.03 | 98.96 | 96.40 | 98.83 | 97.86 | 96.92 |
| Z3 | 90.71 | 98.05 | 97.83 | 98.16 | 92.19 | 96.53 | 98.21 | 96.17 | 99.53 | 99.05 | 96.64 |
| ZALL | 92.17 | 99.41 | 98.45 | 98.87 | 94.73 | 98.01 | 99.12 | 98.08 | 99.69 | 99.72 | 97.83 |
| | | | | Image dimensions 60 × 60 | | | | | | | |
| Z1 | 93.69 | 99.54 | 99.25 | 99.52 | 97.81 | 98.74 | 99.66 | 99.07 | 99.56 | 99.37 | 98.62 |
| Z2 | 95.72 | 99.54 | 99.46 | 99.28 | 96.31 | 99.11 | 99.73 | 98.99 | 99.12 | 99.63 | 98.69 |
| Z3 | 97.43 | 99.20 | 98.84 | 99.41 | 81.69 | 98.59 | 98.78 | 98.85 | 99.37 | 99.84 | 97.20 |
| ZALL | 97.59 | 99.77 | 99.56 | 99.60 | 97.93 | 99.26 | 99.75 | 99.46 | 99.84 | 99.84 | 99.26 |

**Table 7.** Ten-class classification of compressed images.

| Class | ORI | MD1 | MD2 | MD3 | GA1 | GA2 | GA3 | AG1 | AG2 | AG3 | Overall |
|---|---|---|---|---|---|---|---|---|---|---|---|
| | | | | Image dimensions 30 × 30 JQF = 75 | | | | | | | |
| Z1 | 69.52 | 78.99 | 86.11 | 88.11 | 57.22 | 56.06 | 85.18 | 63.46 | 92.61 | 97.34 | 77.46 |
| Z2 | 66.48 | 79.70 | 81.43 | 90.85 | 63.93 | 62.47 | 84.91 | 59.99 | 91.72 | 96.49 | 77.80 |
| Z3 | 73.74 | 80.09 | 84.97 | 89.41 | 53.83 | 60.57 | 86.35 | 60.50 | 89.83 | 95.91 | 77.52 |
| ZALL | 70.11 | 80.28 | 84.41 | 90.79 | 63.14 | 61.31 | 87.25 | 64.93 | 92.83 | 97.49 | 79.25 |
| | | | | Image dimensions 60 × 60 JQF = 75 | | | | | | | |
| Z1 | 78.19 | 88.95 | 94.54 | 94.85 | 76.74 | 87.61 | 96.92 | 62.59 | 92.39 | 98.71 | 87.15 |
| Z2 | 80.57 | 91.54 | 93.43 | 97.39 | 72.09 | 77.87 | 95.01 | 73.54 | 97.45 | 99.46 | 87.83 |
| Z3 | 80.66 | 93.39 | 94.25 | 97.36 | 72.99 | 67.56 | 85.97 | 82.70 | 97.67 | 98.89 | 87.14 |
| ZALL | 80.97 | 92.42 | 94.35 | 97.69 | 78.53 | 78.40 | 94.96 | 77.02 | 96.85 | 99.35 | 89.05 |

In Table 2, the classification accuracy is shown between non-filtered (ORI) and filtered images, i.e., between two classes. The classification accuracy is more than 99%, except for the ORI-GA1 pair (84.72%) for 30 × 30 uncompressed images. Images with dimensions 60 × 60 follow a similar pattern; except for the ORI-GA1 pair (91.87%), the classification accuracy is greater than 99%. Therefore, the results for the 60 × 60 uncompressed images are not displayed in the table.

In Table 3, outcomes for compressed images with JQF = 75 and dimensions of 30 × 30 or 60 × 60 are exhibited. Gaussian-filtered images GA1, GA2, and GA3 are created using the 0.5, 1.0, and 1.5 standard deviations; due to this, ORI-GA1 has the least detection accuracy. ZALL provides 93.06% classification accuracy on ORI-MD1 pairs on the 30 × 30 images. In the case of the 3 × 3 filter, the ORI-AG1 pair has the highest detection accuracy, i.e., 96.49%, compared to pairs ORI-GA1 and ORI-MD1. For low-pass filters of size 5 × 5, the detection accuracies are 96.26%, 99.39%, and 98.87% for ORI-GA2, ORI-AG2, and ORI-MD2, respectively. The detection accuracies for low-pass filters of size 7 × 7 are nearly 99%.

In Table 4, detection accuracies are demonstrated for four-class classification in the case of the Gaussian filter using the proposed scheme on uncompressed images. ORI, GA1, GA2, and GA3 are the four classes. Detection accuracies are displayed for individuals and four classes (overall). Most of the misclassification is in the ORI and GA1 classes. However, the ZALL delivers 95.71% and 98.85% overall classification accuracy for the 30 × 30 and 60 × 60 images, respectively. The median- and averaging-filtered results are not included in Table 4. Regarding median and averaging filtering, ZALL delivers 98.89% and 99.93%

overall classification accuracy, respectively, for $30 \times 30$ images. The general classification accuracy of median and averaging filtering is more than 99.61% and 99.97%, respectively, for the $60 \times 60$ images.

In Table 5, four-class classification was executed using the proposed scheme on compressed images of JQF = 75. Overall detection accuracy on test images is 81.62% and 89.25% in the case of Gaussian-filtered images for the $30 \times 30$ and $60 \times 60$ images, respectively. The overall detection accuracy for averaging-filtered images is 96.85% for the $30 \times 30$ images and 99.02% for the $60 \times 60$ images. Overall classification accuracy for median-filtered images is 92.09% for the $30 \times 30$ images and 97.81% for the $60 \times 60$ images.

ZALL provides the highest detection accuracy, as shown in Table 6, on uncompressed images of dimensions $30 \times 30$ or $60 \times 60$ for ten-class classification. In ten classes, one is ORI class; three are Gaussian-filtered, three are averaging-filtered, and three are median-filtered class. Images were classified simultaneously using a tri-layered neural network classifier. The overall detection accuracies are 97.83% and 99.26% on the $30 \times 30$ and $60 \times 60$ images, respectively. On observing the misclassified images, the number of misclassified images is highest between ORI and GA1, as the GA1 variance is only 0.5.

Table 7 displays results for ten-class classification on compressed images (JQF = 75) of dimensions $30 \times 30$ or $60 \times 60$. ZALL also delivers the maximum number of correct predictions. The average detection accuracies are 79.25% and 89.05% on the $30 \times 30$ and $60 \times 60$ images, respectively. There is approximately a ten percent difference between detection accuracies for the $30 \times 30$ and $60 \times 60$ images. The environment is challenging, as the same filter with a different configuration is also included with other filters.

The proposed scheme was compared with some other prevalent schemes [8,10,11,13–15] for ten-class classification overall detection accuracy. The schemes [8,10,11] rely on manual feature fetching. The scheme in [8] is well known as GLF. The features are fetched by co-occurrence and correlation within the pixels. The schemes [10,11] exploit the autoregressive model. The results of deep neural network schemes are far better than manual schemes, as apparent from Table 8. The techniques [13–15] originated using a deep learning network. Chen et al. [13] utilized the five convolutional layers and considered the residual images and pooling layers. Liu et al. [14] considered the frequency domain, max-pooling layers, and only two convolutional layers. Zhang et al. [15] used squeeze and excitation blocks. In the proposed scheme, multiple measures are applied to improve the detection accuracy. The suggested design adds the global variance pooling layer to enhance the deep neural network's detection capabilities. The spatial and discrete Fourier transform domain features are concatenated to enrich the feature array. Feature arrays from three trained networks are combined, and a leaky ReLU layer is used instead of a traditional ReLU layer to improve the outcomes. Therefore, the results of the proposed scheme are better. The proposed scheme results are superior to Zhang et al. [15] by at least three percent. The presented scheme results are also much higher than results from other schemes [8,10,11,13,14].

**Table 8.** Proposed scheme compared with other schemes.

| Image Dimensions | Kang et al. [10] | Chen et al. [8] | Yang et al. [11] | Chen et al. [13] | Liu et al. [14] | Zhang et al. [15] | Proposed Scheme |
|---|---|---|---|---|---|---|---|
| Uncompressed images | | | | | | | |
| $30 \times 30$ | 58.28 | 67.48 | 69.39 | 76.85 | 81.79 | 94.16 | 97.83 |
| $60 \times 60$ | 68.88 | 83.90 | 78.34 | 83.97 | 91.87 | 96.33 | 99.26 |
| Compressed images with JQF = 75 | | | | | | | |
| $30 \times 30$ | 51.31 | 55.72 | 53.33 | 58.39 | 68.44 | 75.37 | 79.25 |
| $60 \times 60$ | 62.22 | 65.60 | 74.10 | 70.39 | 68.72 | 83.21 | 89.05 |

## 4. Conclusions

Image manipulation has become increasingly prevalent on web platforms, with various operations used to generate fake images, including low-pass filtering. In this paper, a hybrid scheme has been proposed to detect low-pass filtering in small digital images. The proposed approach used a deep neural network with three different solvers and a leaky ReLU layer to improve classification accuracy. We also introduced a global variance pooling layer to boost performance. In addition, frequency domain features were utilized to generate a more robust feature array. A tri-layered neural network classifier was used to classify Gaussian, median, and averaging filters with diverse parameters. The experimental results of the proposed scheme are superior to several other existing techniques. The proposed scheme acquired 81.62%, 96.85%, and 92.05% detection accuracies for Gaussian-, averaging-, and median-filtered JPEG-compressed $30 \times 30$ images on four-class classification.

**Author Contributions:** Conceptualization, S.A. and K.-H.J.; software, S.A.; validation, K.-H.J.; formal analysis, S.A.; investigation, S.A.; resources, S.A.; data curation, K.-H.J.; writing—original draft preparation, S.A.; writing—review and editing, K.-H.J.; visualization, S.A.; supervision, K.-H.J.; project administration, K.-H.J.; funding acquisition, K.-H.J. All authors have read and agreed to the published version of the manuscript.

**Funding:** This work was supported by a Research Grant from Andong National University.

**Data Availability Statement:** The datasets used in this paper are publicly available, and their links are provided in the reference section.

**Conflicts of Interest:** The authors declare no conflict of interest.

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
