# Peer review of "Enhancing Low-Pass Filtering Detection on Small Digital Images Using Hybrid Deep Learning"

_electronics, doi:10.3390/electronics12122637_

Round 1

Reviewer 1 Report

In this paper, a novel scheme is proposed to detect the use of low-pass filters in image processing. A new convolutional neural network with a reasonable size is designed to identify three types of low-pass filters. The learning experiences of the three solvers are combined to enhance the detection ability of the proposed approach. The paper is interesting but it is not well-written and has many drawbacks:

1. The paper is not well-structured. The authors presented different techniques and approaches and mixed them in confused way. There are no any conclusions about advantages and disadvantages of each technique. The difference between schemes for each task is not clear. Figure 3 is not clear. 

2. The results of the paper are not clearly presented. Part 3 is practically not clear.

3. How was accuracy calculated? This point is absolutely not clear.

4. The paper needs part Discussion.

5. Figure 9 is not clear. What is difference between MD 1-3 or AG 1-3 or GA 1-3?

6. Why do the authors don't present values of SSIM metrics at the end of the paper together with the tables in Part 3?

7. The organization of the paper should be added to Introduction. 

Minor editing of English language required. Sometimes the authors use too long sentences.

Author Response

Thank you for your kind comments. Please check the attached file.

Reviewer 2 Report

Review on the paper entitled “Low-pass filtering detection on small digital images using a deep hybrid learning” by authors Saurabh Agarwal and Ki-Hyun Jung submitted to the journal MDPI Electronics

Nowadays, image manipulation is one of the most used techniques dealing with digital images. One the one hand, many of those are done almost automatically in various intelligent systems. On the other hand,  however, some of those can easily be used to produce fake images and manipulate the people. For this reason, the topic of the paper is interesting and important: how one can detect fake images that are manipulated with some techniques. The paper focuses on low-pass filtering as manipulation techniques. The authors already have some related results that they have published in [4].

The paper is written in the usual style with a good English usage. The shown results are promising, however, before publishing them, I recommend some extensions of the study according to the following comments.

At the comparison with other techniques, some hints should be given why the proposed model is more efficient than the others, what differences could result these improvements.

The experiment should be expanded by using images that are manipulated with other methods and/or with some other methods and low-pass filtering (maybe in various orders). It should be examined what other type of manipulation techniques may fool the system, and also, how the system behaves if some low-pass filtered images are further manipulated. Can this effect be still shown with a high accuracy?

Some other comments:

Many systems are called in the paper “deep networks” sometimes with 3 or 5 layers. I am not sure, if the “deep” terminology is adequate for all these neural networks. Sometimes, also the question is arisen, why 9 convolutional layers are used, why not less or more?

Maybe it is better to use other sign for multiplication than * in mathematical formulae, e.g., in page 4.

“after applying filtering filtered images” in page 8.

There is a table that is broken by a page break (pages 8 and 9).

The language and grammar use is generally nice. Only a small issue, repetition is “after applying filtering filtered images” in page 8.

Author Response

(The authors gave the same response as above.)

Reviewer 3 Report

This paper proposed CNN with a reasonable size designed to identify three types of low-pass filters.

There are several comments from reviewers to improve paper quality.

1. On lines 114-115, the paper said, "A leaky ReLU layer is used instead of the conventional ReLU layer in the proposed deep network to improve the detection capability." please prove it on the experimental result to show the marginal result between leaky ReLU and ReLU. The deductive should be proven.

1. Please explain the abbreviation, such as SC.

2. Please bold the best result on each table for easy reading.

Author Response

(The authors gave the same response as above.)

Round 2

Reviewer 3 Report

Thank you for the response.

All comments have been addressed in the current manuscript.